# Examination of Construct Validity and Criterion-Related Validity of the German Motor Test in Egyptian Schoolchildren

**DOI:** 10.3390/ijerph18168341

**Published:** 2021-08-06

**Authors:** Osama Abdelkarim, Julian Fritsch, Darko Jekauc, Klaus Bös

**Affiliations:** 1Institute of Sports and Sports Science, Karlsruhe Institute of Technology, 76131 Karlsruhe, Germany; julian.fritsch@kit.edu (J.F.); darko.jekauc@kit.edu (D.J.); klaus.boes@kit.edu (K.B.); 2Faculty of Physical Education, Assiut University, Assiut 71515, Egypt

**Keywords:** physical fitness, construct validity, schoolchildren

## Abstract

Physical fitness is an indicator for children’s public health status. Therefore, the aim of this study was to examine the construct validity and the criterion-related validity of the German motor test (GMT) in Egyptian schoolchildren. A cross-sectional study was conducted with a total of 931 children aged 6 to 11 years (age: 9.1 ± 1.7 years) with 484 (52%) males and 447 (48%) females in grades one to five in Assiut city. The children’s physical fitness data were collected using GMT. GMT is designed to measure five health-related physical fitness components including speed, strength, coordination, endurance, and flexibility of children aged 6 to 18 years. The anthropometric data were collected based on three indicators: body height, body weight, and BMI. A confirmatory factor analysis was conducted with IBM SPSS AMOS 26.0 using full-information maximum likelihood. The results indicated an adequate fit (χ^2^ = 112.3, df = 20; *p* < 0.01; CFI = 0.956; RMSEA = 0.07). The χ^2^-statistic showed significant results, and the values for CFI and RMSEA showed a good fit. All loadings of the manifest variables on the first-order latent factors as well as loadings of the first-order latent factors on the second-order superordinate factor were significant. The results also showed strong construct validity in the components of conditioning abilities and moderate construct validity in the components of coordinative abilities. GMT proved to be a valid method and could be widely used on large-scale studies for health-related fitness monitoring in the Egyptian population.

## 1. Introduction

Physical fitness (PF) is classified as a public health indicator affecting physical, mental, and psychological aspects [1,2,3,4,5]. In addition, PF is also suggested to play a vital role in brain functions and learning performance [6,7,8]. Thus, to effectively combat numerous public health problems, especially childhood obesity, there is a critical need for increasing PF levels among children [9]. In this context, PF should be ideally promoted at a young age in order to avoid long-lasting health problems and possibly improve cognitive functions and mental health [10].

A low level of PF and insufficient physical activity are associated with greater somatic and psychological problems [1,2]. Therefore, increased levels of PF, specifically muscular strength, could have significant benefits for the psychological health of overweight/obese children [11]. In addition, high PF levels are associated with less unfavorable body composition among children with elevated school stress [12]. Moreover, PF is also related to social health, such that the school environment is more conducive to the development of PF for promoting the students’ social health [5].

Indeed, PF is usually determined in school-aged children using health-related PF batteries (e.g., field tests) [13]. Therefore, having good criteria for collecting PF data among children and youths is useful for identifying problems or optimizing performances in order to provide intervention programs that develop children’s status [14,15,16]. Indeed, the assessment of PF became a necessary topic in epidemiological studies, since a reduction in PF is directly associated with the incidence of obesity, coronary heart diseases, diabetes, and hypertension in adults [17,18,19].

### Motor Performance Abilities (MPA)

There is a large body of historical work on the differentiation of motor abilities [20,21,22,23,24,25,26,27]. All approaches are based on the idea that the motor system is a complex, multidimensional construct that cannot be adequately described by one single characteristic. Differentiations of motor abilities are mostly based on the assumption of so-called basic qualities [28] or major forms of physical performance [29] and allow for a sufficiently accurate initial diagnosis and orientation for controlling the exercise load in physical education in the basic training of competitive sports but also in health and rehabilitation sports. However, differentiations according to ability categories are not sufficiently precise for performance explanations and prognoses, for training control at a high performance level, for sport-specific description models, or for disease-specific questions in rehabilitation. Here, diagnoses require a more process- and function-oriented approach with the help of sports-medicine or biomechanical measurement methods.

According to Bös and Mechling [30], MPA is on a first level differentiated according to the poles of energy and information into physical (energetic) and coordinative (information-oriented) abilities. On a second level, there is a breakdown into the much-cited “basic motor qualities” of endurance, strength, speed, coordination, and flexibility. The assignment of endurance and strength abilities to the energetically determined functional processes results from the distinction between the cardiovascular system and the skeletal muscles as central systems of energy production and energy transport in the human organism. The extent, the mass, and the structure of skeletal muscle are considered prerequisites for strength performance. The performance of the cardiovascular system represents determining and limiting variables for endurance performance.

In this context, the German motor test (GMT) [31] was developed as an objective tool to measure a complete fitness profile involving speed, endurance, strength, coordination under precision demands, coordination under time pressure, and flexibility [32,33]. The test battery was designed to be easily used in sports gym for physical examination. The content-related validity of all test items was consistently rated as good in terms of significance and feasibility based on expert ratings.

In Egypt, the health-related fitness components of schoolchildren were not sufficiently studied in recent decades, which negatively affects any intervention strategies, plans, or programs. Currently, it is useful for educational settings, parents, clinicians, and sports organizations to have valid and reliable health-related fitness data about children. In this context, this test could be used in Egypt as a data-collection tool for identifying problems or excellent performance in the field of public health, talent identification, exercise pediatric, and epidemiological studies [9,11,14,15,16].

Indeed, providing a valid and reliable tool for measuring health-related fitness components is very helpful for long-term and sustainable development of sport and health ecosystems in Egypt. Therefore, the aim of this study was to examine the construct validity and the criterion-related validity of the German motor test in Egyptian children aged 6 to 11 years.

## 2. Materials and Methods

### 2.1. Study Design and Sampling

A cross-sectional study was conducted between 2014 and 2017 at 13 public primary schools in the city of Assiut, which is the largest town in upper Egypt and is located about 234 miles south of Cairo. The size of the primary-school student population aged 6 to 11 years in Assiut is about 76,334 students. The total number of public schools is 69, distributed in 7 districts. The final sample was randomly selected from the chosen schools and consisted of 931 children aged 6 to 11 years (age: 9.1 ± 1.7 years) with 484 (52%) males and 447 (48%) females in grades 1 to 5 of primary schools in Assiut. 

Precisely, the test development was based on an international expert survey involving 40 selected fitness experts in 25 European countries who were asked about the relevance of the test contents and requirements in sport-motor tests with respect to MPA documentation [34]. Subsequently, 13 experts evaluated the significance and the practicality of the study exercises on a scale of 1 (very good) to 5 (very bad). The evaluations in both regions were found to be within a good range (M_Significance_ = 1.9; M_Practicality_ = 1.7). To determine test–retest reliability, the motor tests were performed twice within 4 days on the same children using the same test situation and the same study investigator. All in all, there were good test–retest reliability coefficients (R_min_ = 0.74 to R_max_ = 0.96).

### 2.2. Data Collection 

#### 2.2.1. Anthropometric Characteristics Data 

Assessment of the anthropometric characteristics of the children was based on three indicators: body height, body weight, and BMI. The instruments were calibrated according to the standard preparation prior to measurement. Measurements were taken with schoolchildren wearing light clothing and no shoes. Body weight was measured with a beam balance to the nearest 0.1 kg. Body height was measured with a stadiometer to the nearest 0.5 cm. BMI was defined as the ratio of body weight to body height squared, expressed in kg/m. Subjects were classified as underweight, normal weight, and overweight (i.e., overweight and obese) according to the published standards by the International Obesity Task Force based on age and sex difference characteristics [35].

#### 2.2.2. Physical Fitness Data

The German motor test (GMT) was used to measure five health related-physical fitness components of children aged 6 to 18 years [36]. To determine test–retest reliability in the Egyptian sample, the test was performed twice within 7 days with the same children using the same test situation and the same study investigator. Good test–retest reliability coefficients were obtained (R-values between 0.68 and 0.94). The test items were described and performed according to Lämmle et al. [37] and Abdelkarim et al. [38] as the following:

Speed: The 20 m sprint test is used to measure speed ability. The child must cover a distance of 20 m in as short a time as possible in two trials; the best trial is evaluated. The time required for the sprint is measured to the nearest tenth of a second using a stopwatch (the start is from a standing position).

Coordination: Balancing backwards (BB) is used to measure coordination under precision demands. The child must walk backwards over three beams of approximately the same length (300 cm) but different widths (6 cm, 4.5 cm, and 3 cm) in two valid trials while maintaining balance. The goal is to stay on each of the beams, i.e., not miss, during the course of two valid trials. A total of six successful trials are evaluated. The number of steps taken while walking backwards is counted. The variable used for the analysis is the sum of the steps taken during all six trials while walking backwards. Side jumping (JS) is used to measure whole-body coordination under time pressure, speed, and muscular endurance of the lower extremities. The child must jump sideways across the center line of the carpet mat with both legs at the same time as fast as possible without exceeding the given field size (50–100 cm). Two trials of 15 s each are performed. The recovery time between the trials is 1 min. The number of jumps made during the two trials is evaluated, and the average of the two trials is analyzed.

Strength: Push-ups (PU) and sit-ups (SU) are used to measure dynamic muscular endurance of the upper extremities and the abdominal muscles, respectively. The child is asked to perform as many push-ups or sit-ups as possible in two trials of 40 s each. The starting position of the push-ups is the lying position with hands clasped behind the back. The second position is the raised position of the standard push-up (i.e., with arms extended). In the third position, one hand touches the top of the hand of the supporting arm before returning to the starting position. The average of the two attempts is used. The standing long jump (SLJ) test is used to measure the jumping strength and the spring strength of the leg muscles. The test person jumps with both legs and tries to reach the greatest possible distance. The propulsion may only be increased by swinging the arms, but it is not allowed to reach back with one or both hands. The distance (in cm) from the starting line to the heel of the foot behind when landing is measured. The best of two jumps is used for analysis.

Endurance: A six-minute running test is used to measure aerobic endurance. The child is asked to run around a volleyball court as many times as possible within six minutes. The measurement for each child is the distance in meters covered within six minutes. The length of the path is the number of laps (1 lap = 54 m) plus the distance covered in the last lap.

Flexibility: The stand-and-reach (SR) test is used to measure trunk flexibility and the elasticity of back and leg muscles. The test person stands on a wooden box and slowly bends forward at the waist. The arms and the hands must reach down as far as possible with the legs extended. The better of two trials is noted in centimeters.

### 2.3. Statistical Analysis

A confirmatory factor analysis was conducted with IBM SPSS AMOS 26. (IBM Corp.: Armonk, NY, USA) [39] using full-information maximum likelihood, which has the advantage that, when models with missing values are computed, the estimates are less biased than when classical methods such as listwise deletion, pairwise deletion, or mean imputation are used to handle missing values [40]. A five-factor structure with a global factor of physical fitness was assessed. The assessment of global goodness-of-fit was based on several fit indices. First, a non-significant p-value in the χ^2^-statistic indicates a good model fit [41]. However, this test depends on the sample size, and even minor differences between the implied model and the observed covariance matrix led to significant results [42].

Second, the comparative fit index (CFI) shows the relative fit improvement by comparing the proposed model with the baseline model. Cut-off values for CFI are desirable above 0.95 and adequate above 0.90 [43]. Third, the root mean square error of approximation (RMSEA) describes the error of approximation in the population. RMSEA values are adequate below 0.08 and desirable below 0.05 [44]. To examine criterion-related validity, the relationship between physical fitness and children’s BMI was assessed using bivariate correlations. For that purpose, a dummy variable was created to compare children who are classified as overweight or obese with those not classified as such [35]. Separate models were estimated for each calculation of bivariate correlations between the dummy variable and the latent variables.

## 3. Results

### 3.1. Descriptive Statistics 

Table 1 contains raw score means, standard deviations, and correlations between all test indicators. The correlation coefficient between test items was easily demonstrated. The highest correlation coefficient values were shown between the 20 m sprint test and the long jump test (R = −0.67). On the other hand, the lowest correlation coefficient was demonstrated between the 6 min run and flexibility (R = 0.08), which is classified as a passive system of energy transfer. However, significant correlations between the 6 min run test and the test items related to strength ability were shown (R_pushup_ = 0.30, R_situp_ = 0.42, and R_longjump_ = 0.46). There was also a strong correlation between the test item for the 20 m sprint and the test items for push-ups (R = 0.39), sit-ups in 40 s (R = 0.50), and the long jump (R = −0.67), respectively. The test items measuring coordination ability (jumping sideways, balancing backwards) showed a moderate significant correlation with the test items measuring sprint and strength ability. High correlations were shown between the jumping sideways test item and the 20 m sprint (R = −0.51), the long jump (R = 0.47), the push-ups (R = 0.43), and the sit-ups in 40 s (R = 0.40).

### 3.2. Construct Validity 

Bös and Mechling’s model [30] was presented in (Figure 1) as a structural equation model. The dimensions of endurance, coordination under time pressure, coordination with precision demands, and flexibility were operationalized with one item each. The strength dimension included four indicators. The superordinate dimension was motor performance ability (see Figure 2). The results of the model indicated an adequate fit (χ^2^ = 112.3, df = 20, *p* < 0.01; CFI = 0.956; RMSEA = 0.07). Although the χ^2^-statistic showed significant results, the values of CFI and RMSEA showed a good fit. All loadings of the manifest variables on the first-order latent factors and the loadings of the first-order latent factors on the second-order superordinate factor were significant.

### 3.3. Criterion-Related Validity

Regarding BMI, we used a variable comparing children classified as overweight or obese with those not classified as such. This variable correlated significantly with endurance (R = −0.22, Z =− 4.58, *p* < 0.01), coordination with precision demands (R = −0.40, Z = −5.98, *p* < 0.01), and overall motor performance ability (R = −0.16, Z = −4.66, *p* < 0.01) but not with flexibility (R = 0.04, Z = 0.34, *p* = 0.73) or strength (R = −0.04, Z = −1.51, *p* = 0.13). Interestingly, contrary to our expectations, the results showed that time pressure was positively correlated with BMI (R = 0.17, Z = 4.19, *p* < 0.01).

## 4. Discussion

Construct validity and criterion-related validity of the German motor test (GMT) were studied in Egyptian schoolchildren. The results of the confirmatory factorial analysis showed a good fit of Bös and Mechling’s model [30] for the structure of motor performance abilities with good values for CFI and RMSEA. The criterion validity coefficient was acceptable for the majority of the test items. GMT showed strong construct validity in the components of energetically determined (conditioning) abilities including cardiorespiratory capacity (endurance), muscle strength, and speed. However, the test showed moderate construct validity in the components of information-oriented (coordinative) abilities including coordination with precision demands and coordination under time pressure.

The results showed a good construct validity with high significant values for the loading of the test items. The second-order factor was based on the five first-order factors of endurance, strength, coordination under time pressure, coordination with precision demands, and flexibility. The results suggested comparable results to other studies using the same test battery [36,45]. In addition, the current results confirmed the previous results of another test battery consisting of a combination of speed, endurance, strength, coordination, and flexibility that were shown to be valid, functional, and easy to administer for measuring children’s physical fitness in different European populations in the same age groups [16,46].

The results also showed high loadings for the test items for strength, coordination with precision demands, and coordination under time pressure. These high loadings point to the potential of these test items as indicators of performance level [37]. In contrast, the loading of flexibility showed the lowest loading value. This confirmed the assumption that flexibility is a rather independent dimension (passive system of energy transfer). Moreover, flexibility shows a general deterioration in performance with age growing in both genders, especially in girls [47].

Indeed, the validity of the German motor test (GMT) in Egyptian children confirms the importance of measuring the levels of PF based on quantitative measurements such as anthropometric data and health-related fitness batteries [48,49]. These types of measurements are more practical, motivating, and provide an accurate overview of fitness levels, especially in the child population. In addition, the association between physical fitness and body mass index was shown to provide an indication of test validity, especially in prepubertal school children [38,50]. However, greater effort and logistical support are needed for data collection in large-scale studies [17,51,52].

This study provides an economical and objective data collection tool to increase the possibility of national representative studies for health-related physical fitness in Egypt. The tests included in the tool can help to provide an overview about Egyptian children’s rate of involvement in physical activity. PF provides objective data which could predict the rate of participation in physical activity. Guthold et al. [53] point out that nationally representative data for physical activity using scientific measurements, such as accelerometers, are only available for high-income countries. Low-income countries, mainly in the Middle East and North Africa, had a very low proportion of available data with the estimated overall percentage of insufficient physical activity reaching 32%. Here, the WHO recommendation on physical activity and sedentary behavior should be strongly considered to achieve benefits in children and adolescents for improved physical fitness (cardiorespiratory and muscular fitness), cardiometabolic health (blood pressure, glucose, and insulin resistance), bone health, cognitive outcomes (academic performance, executive function), mental health (reduced symptoms of depression), and reduced obesity [54].

## 5. Conclusions

The German motor test (GMT) was shown to be a valid method for measuring PF in children in Egypt. This valid tool for data collection opens a large window for researchers to use in large-scale studies monitoring health-related fitness components in the fields of epidemiology, talent identification, and health-related educational studies. However, larger and representative samples are needed to establish reference standards in Egyptian children to correctly interpret the results of such tests by generating sex- and age-specific normative percentile values to be available for comparative studies and to establish a national database and fitness profile for the Egyptian population.

## Figures and Tables

**Figure 1 ijerph-18-08341-f001:**
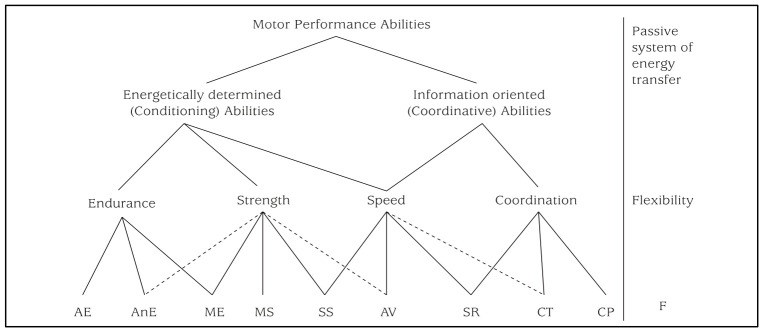
Motor performance abilities [30]. AE = aerobic endurance; AnE = anaerobic endurance; ME = muscular endurance; MS = maximum strength; SS = speed strength; AV = action velocity; SR = speed of response; CT = coordination under time pressure; CP = coordination with precision demands; F = flexibility.

**Figure 2 ijerph-18-08341-f002:**
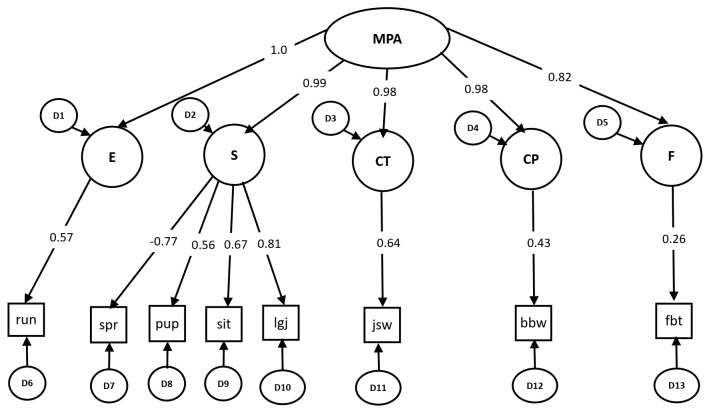
MPA = motor performance ability; E = endurance; S = strength; CT = coordination under time pressure; CP = coordination with precision demands; F = flexibility; run = 6 min run; spr = 20 m sprint; pup = push-ups; sit = sit-ups in 40 s; lgj = long jump; jsw = jumping sideways; bbw = balancing backwards; fbt = forward bending of the trunk.

**Table 1 ijerph-18-08341-t001:** Descriptive statistics.

	M	SD	Run	Spr	Pup	Sit	Lgj	Jsw	Bbw	Fbt
**Run**	7840.19	1690.42	1							
**Spr**	40.66	00.63	−0.43 **	1						
**Pup**	90.77	40.64	0.30 **	−0.39 **	1					
**Sit**	140.54	70.44	0.42 **	−0.50 **	0.46 **	1				
**Lgj**	1110.12	260.06	0.46 **	−0.67 **	0.42 **	0.54 **	1			
**Jsw**	230.16	70.48	0.36 **	−0.51 **	0.43 **	0.40 **	0.47 **	1		
**Bbw**	260.66	110.04	0.26 **	−0.27 **	0.25 **	0.27 **	0.38 **	0.26 **	1	
**Fbt**	−20.90	60.76	0.08 *	−0.12 **	0.15 **	0.16 **	0.18 **	0.13 **	0.20 **	1

* *p* < 0.05; ** *p* < 0.01; M = mean; SD = standard deviation; run = 6-min run; spr = 20-m-sprint; pup = push-ups; sit = sit-ups in 40 s; lgj = long jump; jsw = jumping sideways; bbw = balancing backwards; fbt = forward bending.

## Data Availability

The data presented in this study are available on request from the corresponding author. The data are not publicly available due to technical issue.

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
