# Peer review of "Examination of Construct Validity and Criterion-Related Validity of the German Motor Test in Egyptian Schoolchildren"

_ijerph, 2021, doi:10.3390/ijerph18168341_

Round 1

Reviewer 1 Report

Dear Authors,

although that you submitted an interesting contribution to the body of health-related studies on physical fitness and motor competence of elementary school children, major revisions are needed in regard to several flaws of the manuscript (MS). 

General remarks:

- I suggest to use the terms “Physical fitness (PF)” tests/components and “Motor competence (MC)” tests/components instead of “conditioning abilities” and coordination abilities” throughout the MS. See, for example:

Cattuzzo, M.T.; Dos Santos Henrique, R.; Ré, A.H.N.; de Oliveira, I.S.; Melo, B.M.; de Sousa Moura, M.; de Araújo, R.C.; Stodden, D.F. Motor competence and health related physical fitness in youth: A systematic review. J. Sci. Med. Sport 2016, 19, 123–129, doi:10.1016/j.jsams.2014.12.004.

Lima, R.A.; Pfeiffer, K.; Larsen, L.R.; Bugge, A.; Moller, N.C.; Anderson, L.B.; Stodden, D.F. Physical Activity and Motor Competence Present a Positive Reciprocal Longitudinal Relationship Across Childhood and Early Adolescence. Journal of Physical Activity and Health 2017, 14, 440–447, doi:10.1123/jpah.2016-0473.

- Language should be revised thoroughly throughout the MS.

- Please, comment on the large age range in the sample, and the possible effects on the study results. Please, continue this reasoning in the Discussion section.

- Please, comment on the pros and cons to summarize the data of boys and girls into one single sample in the study.

Abstract

Language:

- It should be “The children’s physical fitness was assessed using DMT … ”? See also in the following text!

- It should be “widely” instead of “wildly”. See also the Conclusion section.

- “… on large scale …”

Introduction

- As Motor Performance (see Fig. 1) comprises PF and MC abilities (better: components) the relevance of MC should be clarified in the Introduction section (before chapter 1.1)

- Early mortality (ref. 19) does not play a significant role in the context of the study conducted. This remark is gratuitous.

- The reference to “major forms of stress (ref. 29)” is unclear – please explain.

- Strength performance instead of “motor strength ability”.

- What does the author mean by “small rooms”? Is this in line with the ideas of the `inventors` of the DMT ? If so, please, provide a reference.

- Please clarify the acronym “MPA” as it is used for the first time in the MS!

- The chapter “Precisely … coefficients …” should be transferred to the Materials and methods section (2.2.2)

Materials and methods

- Typo: “Components” instead of “componanta”.

Please, comment on the calculation method of MPA (mean or sum of z-values etc.).

Results

- Please, in Chapter 3.1 rephrase the sentence “However, a significant correlation coefficients values …”

- Please, in Chapter 3.1 rephrase the sentence “The strongest significant correlation …”

- Please, correct the term “Figure 1” in the legend to Figure 2.

- In Chapter 3.2 it should be “result” instead of “results”.

- Is the positive direction of the correlation between overweight and coordination under time pressure (R = .17) correct?

Discussion

- Please, rephrase the sentence “Here, the results show that the model fits …”

- Typo: “batteries” instead of “batirries”. Please, rephrase the whole sentence.

- The parenthesis in the sentence “The results also indicated …” is unclear.

- Typo: “shown” instead of “showen”.

- “Please, rephrase the sentence “Actually, using this valid and available (?) test …”.

- The meaning of the sentence “PF provides an objective data which is could predict the rat of participating in physical activity” is totally unclear. In my eyes the whole chapter should be rephrased.

Conclusion

- The idea that the DMT could be used for “epidemiological, talent identification, and health-related educational studies” should be introduced in the Introduction section already, and should be supported by references.

Author Response

Dear Reviewer_1,

Thank you very much for your efforts!

General remarks:

- I suggest to use the terms “Physical fitness (PF)” tests/components and “Motor competence (MC)” tests/components instead of “conditioning abilities” and coordination abilities” throughout the MS. See, for example:

Cattuzzo, M.T.; Dos Santos Henrique, R.; Ré, A.H.N.; de Oliveira, I.S.; Melo, B.M.; de Sousa Moura, M.; de Araújo, R.C.; Stodden, D.F. Motor competence and health related physical fitness in youth: A systematic review. J. Sci. Med. Sport 2016, 19, 123–129, doi:10.1016/j.jsams.2014.12.004.

Lima, R.A.; Pfeiffer, K.; Larsen, L.R.; Bugge, A.; Moller, N.C.; Anderson, L.B.; Stodden, D.F. Physical Activity and Motor Competence Present a Positive Reciprocal Longitudinal Relationship Across Childhood and Early Adolescence. Journal of Physical Activity and Health 2017, 14, 440–447, doi:10.1123/jpah.2016-0473.

I would refer to Lämmle et al. saying that we want to be consistent with previous publications on the model. The terms are determined by the theoretical approach by Bös.

- Language should be revised thoroughly throughout the MS. It has been checked by proofreader.

- Please, comment on the large age range in the sample, and the possible effects on the study results. Please, continue this reasoning in the Discussion section.

- Please, comment on the pros and cons to summarize the data of boys and girls into one single sample in the study.

The age range of the sample including both genders is presenting the primary school age children in Egypt. And we believe that larger study within this age is needed to be more representative sample for the Egyptian population.

Abstract

Language:

- It should be “The children’s physical fitness was assessed using DMT … ”? See also in the following text! Done

- It should be “widely” instead of “wildly”. See also the Conclusion section. Done

- “… on large scale …”. Done

Introduction

- As Motor Performance (see Fig. 1) comprises PF and MC abilities (better: components) the relevance of MC should be clarified in the Introduction section (before chapter 1.1)

- Early mortality (ref. 19) does not play a significant role in the context of the study conducted. This remark is gratuitous. Done

- The reference to “major forms of stress (ref. 29)” is unclear – please explain. It has been changed to “major forms of physical performance”

- Strength performance instead of “motor strength ability”. Done

- What does the author mean by “small rooms”? Is this in line with the ideas of the `inventors` of the DMT ? If so, please, provide a reference. Sports gym instead of small rooms

- Please clarify the acronym “MPA” as it is used for the first time in the MS! Done

- The chapter “Precisely … coefficients …” should be transferred to the Materials and methods section (2.2.2). Done

Materials and methods

- Typo: “Components” instead of “componanta”. Done

Please, comment on the calculation method of MPA (mean or sum of z-values etc.).

Results

- Please, in Chapter 3.1 rephrase the sentence “However, a significant correlation coefficients values …”

The sentence was changed to: However, significant correlations between the 6 min run test and the test items referring to strength ability were shown (Rpushup = .30, Rsitup = .42, and Rlongjump = .46).

- Please, in Chapter 3.1 rephrase the sentence “The strongest significant correlation …”

This sentence was changed to: High correlations were shown between the test item jumping sideways and 20m sprint (R = -.51), long jump (R = .47), push-ups (R = .43), and sit-ups in 40 seconds (R = .40).

- Please, correct the term “Figure 1” in the legend to Figure 2. Done

- In Chapter 3.2 it should be “result” instead of “results”. Done

- Is the positive direction of the correlation between overweight and coordination under time pressure (R = .17) correct?

Thank you for this remark. This is indeed an interesting result. We added a sentence in the results section: “Interestingly, in contrast to our expectations, the results indicate that time pressure correlates positively with BMI (R = .17, Z = 4.19, p <. 01).”

Discussion

- Please, rephrase the sentence “Here, the results show that the model fits …”

This sentenced was changed to: The results indicate comparable results to other studies using the same test battery.

- Typo: “batteries” instead of “batirries”. Please, rephrase the whole sentence. Done

- The parenthesis in the sentence “The results also indicated …” is unclear.

This sentenced was changed to: These high loadings point to the potential of these test items as indicators of performance level.

- Typo: “shown” instead of “showen”. Done

- “Please, rephrase the sentence “Actually, using this valid and available (?) test …”.

The sentence was changed to: The tests included in the tool can help to provide an overview about the engagement rate of physical activity among Egyptian children.

- The meaning of the sentence “PF provides an objective data which is could predict the rat of participating in physical activity” is totally unclear. In my eyes the whole chapter should be rephrased.

The whole chapter has been rephrased.

Conclusion

The idea that the DMT could be used for “epidemiological, talent identification, and health-related educational studies” should be introduced in the Introduction section already, and should be supported by references. 

It is a recommendation since the test is not have been widely used in Egypt, Therefore, we believe that in the future it will be very reliable tool for data collection in different fields of research.

Reviewer 2 Report

  1. In the text, please check the abbreviation of German Motor Test (GMT)
  2. Please, add 2-3 rows as an introduction in “Abstract” to understand the readers what is GMT
  3. Assist or assed? Please, check it in each sentence and change it
  4. “Psychologically, perception of PF as being poor… BMI [1-2]”.  Please, rephrase this sentence.
  5. In “.Figure 1”, remove the full stop
  6. Please, define what is “MPA”
  7. capa- bilities. Please, change it
  8. Please, change “tow”
  9. What do you mean with “componata”?
  10. “In two valid attempts, the child must walk backwards and keep their balance” change child to children because you write “they”.
  11. In Endurance measurement, If the six minutes were ending before a complete lap how do you calculate the measurement value?
  12. In Flexibility test, why do not you use the “Sit and Reach” test? In “Stand and Reach” test maybe there is the threat that the children have their legs in flexion.
  13. Table 1, must be removed from “Statistic analysis” and to put in “Results”
  14. “The strongest significant correlation was shown between Jsw and (Rsprint = -.51, Rpushup = .43, Rsitup = .40, and Rlongjump = .47) respectively” Please, write again the sentence.
  15. “In addition, the current results confirms the previous results of another test batirries…” Please, explain the phrase “test batirries”
  16. In “Results” where is the correlation analysis between BMI, weight, and Height and MPA parameters?
  17. Please, edit the English of your text

Author Response

  1. In the text, please check the abbreviation of German Motor Test (GMT) Done
  2. Please, add 2-3 rows as an introduction in “Abstract” to understand the readers what is GMT. Done
  3. Assist or assed? Please, check it in each sentence and change it Done
  4. “Psychologically, perception of PF as being poor… BMI [1-2]”.  Please, rephrase this sentence.

We edited the sentence in the manuscript.

  1. In “.Figure 1”, remove the full stop Done
  2. Please, define what is “MPA”. Done
  3. capa- bilities. Please, change it. Done
  4. Please, change “tow”. Done
  5. What do you mean with “componata”? Done
  6. “In two valid attempts, the child must walk backwards and keep their balance” change child to children because you write “they”.

Answer: The test description was deleted in reason of duplication percentage.   

  1. In Endurance measurement, If the six minutes were ending before a complete lap how do you calculate the measurement value?

Answer: The number of meters in the final round will be added to the completed rounds.

  1. In Flexibility test, why do not you use the “Sit and Reach” test? In “Stand and Reach” test maybe there is the threat that the children have their legs in flexion.

Answer: Actually, the test was designed and the good criteria were tested based on this performance. Therefore, I applied the test as it is designed.

  1. Table 1, must be removed from “Statistic analysis” and to put in “Results” Done
  2. “The strongest significant correlation was shown between Jsw and (Rsprint = -.51, Rpushup = .43, Rsitup = .40, and Rlongjump = .47) respectively” Please, write again the sentence. Done
  3. “In addition, the current results confirms the previous results of another test batirries…” Please, explain the phrase “test batirries”. It has been corrected.
  4. In “Results” where is the correlation analysis between BMI, weight, and Height and MPA parameters?

This correlation is reported in the section 3.3. criterion-related validity. We edited this section to make it more clear.

  1. Please, edit the English of your text. It has been checked by proofreader.

Round 2

Reviewer 1 Report

Dear authors,

thank you very much for the revisions made.

Author Response

Dear reviewer 1,

Thank you very much for your efforts in reviewing our manuscript.

Best regards,

Osama 

Reviewer 2 Report

I did not found some sentences in "Abstract" as an introduction.

195. "was studied" instead of investigated

219. "Has show" change it

237. "estimate" change it

I do not understand why you deleted the information about GMT in section 2.2.2. It is necessary to show how the measurements conducted. 

Author Response

Dear Reviewer 2,

Point 1: I did not found some sentences in "Abstract" as an introduction.

Response: The sentence “ Physical fitness is an indicator of children’s public health statues. Therefore, “ was added.

Point 2: Line 195. "was studied" instead of investigated.

Response: I changed it.

Point 3: Line 219. "Has show" change it.

Response: I changed it.

Point 4: Line 237. "estimate" change it.

Response: I changed it.

Point 5: I do not understand why you deleted the information about GMT in section 2.2.2. It is necessary to show how the measurements conducted. 

Response: You are completely right, but the test description was deleted from methods part in reason of high duplication percentage was mentioned by the editors of IJIRPH. In addition, the test items were precisely described in 2 references including: Lämmle et al. [37] and Abdelkarim et al. [38].